# An Improved YOLOv5 Model: Application to Mixed Impurities Detection for Walnut Kernels

**DOI:** 10.3390/foods12030624

**Published:** 2023-02-01

**Authors:** Lang Yu, Mengbo Qian, Qiang Chen, Fuxing Sun, Jiaxuan Pan

**Affiliations:** College of Optical Mechanical and Electrical Engineering, Zhejiang A & F University, Hangzhou 311300, China

**Keywords:** YOLOv5, walnut kernels, impurities detection, small object detection

## Abstract

Impurity detection is an important link in the chain of food processing. Taking walnut kernels as an example, it is difficult to accurately detect impurities mixed in walnut kernels before the packaging process. In order to accurately identify the small impurities mixed in walnut kernels, this paper established an improved impurities detection model based on the original YOLOv5 network model. Initially, a small target detection layer was added in the neck part, to improve the detection ability for small impurities, such as broken shells. Secondly, the Tansformer-Encoder (Trans-E) module is proposed to replace some convolution blocks in the original network, which can better capture the global information of the image. Then, the Convolutional Block Attention Module (CBAM) was added to improve the sensitivity of the model to channel features, which make it easy to find the prediction region in dense objects. Finally, the GhostNet module is introduced to make the model lighter and improve the model detection rate. During the test stage, sample photos were randomly chosen to test the model’s efficacy using the training and test set, derived from the walnut database that was previously created. The mean average precision can measure the multi-category recognition accuracy of the model. The test results demonstrate that the mean average precision (*mAP*) of the improved YOLOv5 model reaches 88.9%, which is 6.7% higher than the average accuracy of the original YOLOv5 network, and is also higher than other detection networks. Moreover, the improved YOLOv5 model is significantly better than the original YOLOv5 network in identifying small impurities, and the detection rate is only reduced by 3.9%, which meets the demand of real-time detection of food impurities and provides a technical reference for the detection of small impurities in food.

## 1. Introduction

Food safety has always been a social health issue of great concern to people. Impurity pollution accounts for a large proportion of food pollution and is difficult to avoid [1]. Impurity pollution refers to the presence of other substances or foreign substances in food other than the food itself [2], which will cause physical and psychological harm to consumers. Taking walnut kernels as an example, impurities of walnut kernels can be divided into exogenous impurities (stones, metal parts, and plastic fragments) and endogenous impurities (walnut shells; spoiled walnut kernels) [3], which will seriously affect consumers satisfaction. Therefore, impurity detection is one of the important links to ensure the high quality of nut food [4]. In recent years, many researchers have tried to use imaging detection technology to detect impurities in food, including X imaging technology, terahertz detection technology, spectral detection analysis technology, machine vision detection technology, etc.

In the field of machine vision detection, object detection methods based on deep learning are developing rapidly. According to the nature of the algorithm stages, the current mainstream algorithms can be divided into two categories: one of them is the two-stage algorithm R-CNN series, and the representative algorithms include R-CNN, SPP-Net, and Faster R-CNN [5]. The series of algorithms first generate regions and then classify samples through convolutional neural networks. Xie et al. used the Fsater-RCNN + VGG16 neural network model to detect bone impurities in salmon meat [6]. Wang et al. used the Faster RCNN ResNet101 for detecting the potato surface defects and verified the high recognition accuracy of the model [7]. The point of the R-CNN series of algorithms is that the detection accuracy is high, but the detection speed is defective. The other type is a single-stage algorithm, and the representative algorithm is the YOLO series. The YOLO (You Only Look Once) algorithm directly inputs the entire image into the model network and returns the classification category and location of the bounding box at the output, so it can extract all features in the image and predict all objects [8,9]. Based on the results of various studies, the YOLOv5 algorithm in the YOLO series has better comprehensive detection ability than other YOLO models due to its accuracy and detection accuracy [10,11,12]. Many researchers have applied the YOLOv5 model or the improved YOLOv5 model to the food safety field for object detection [13]. Jubayer et al. used the YOLOv5 model to detect molds on food surfaces and successfully identified the types of molds on food surfaces [14]. On the basis of the original YOLOv5 network model, Chen et al. added a new involution bottleneck module, which reduced the parameters and calculation amount, and introduced the SE module to improve the sensitivity of the model to channel features, establishing a plant pest identification model [15]. Qi et al. borrowed the human visual attention mechanism and added the squeeze-and-excitation module to the YOLOv5 model to achieve a key feature extraction [16]; the trained network model was evaluated on the tomato virus disease test set, and the accuracy rate reached 91.07%. Han et al. adopted the YOLOv5 model based on the flood filling method to achieve cherry quality detection [17].

This paper takes walnut impurities as the detection target. There is a high requirement for the real-time detection rate of impurities for the fast running speed of the walnut processing line [18]. The YOLOv5 model can maintain a higher detection accuracy, while maintaining a higher detection rate [19], so this paper chooses YOLOv5 as the detection model. However, the original YOLOv5 model is challenging to extract image features of impurities in walnut kernels under complex backgrounds. It is hard to detect small impurities such as broken shells, resulting in a low impurity recognition rate. In order to solve the above problems, we take the pursuit of a balance between detection performance and detection rate as the goal and improve the original YOLOv5 network, so that it can more accurately detect the impurities in the image without losing the detection rate. Firstly, a small target detection layer is added to the neck part to improve the model’s ability to detect small impurities. Secondly, the Tans-E module is proposed to replace some of the convolution blocks in the original network. Thirdly, the CBAM module is added to improve the sensitivity of the model to channel features, which is convenient for finding prediction regions in dense objects. Finally, the GhostNet module is introduced to make the model lighter and improve the model detection rate.

## 2. Materials and Methods

### 2.1. Samples Used in the Experiments

There is uncertainty in walnut processing; thus, we selected random sampling without repetition. From March 2022 to April 2022, we randomly collected about 20 kg of walnut kernels with impurities before the manual sorting process from the walnut processing line of the Nut Fried Goods Base in Longgang Town, Lin’an District. All samples were collected three times. The walnut kernels were mixed with broken walnut shells, unqualified walnut kernels and other impurities. The mixture was divided into 40 groups of samples evenly, according to quality.

### 2.2. Images Acquisition System and Dataset Creation

#### 2.2.1. Images Acquisition System

The image acquisition system has two functions: simulating the walnut processing line and take sample pictures. The system consists of a conveyor belt, aluminum profile, the computer, a camera (D435i from Intel, Santa Clara, CA, USA), a black inspection chamber, and an LED light belt [3]. The conveyor belt is divided into two stages: high speed and low speed. The density of the walnut kernel is changeable by controlling the speed difference of the conveyor belt. The camera is used to capture images with resolutions of 1920 × 1080 pixels. The color model is RGB. The camera is located 400 mm above the second conveyor belt [20]. The black inspection chamber, which was made by diffuse reflection plates, is set to cover the camera. Four equal power light belts (10 W each) are set in the black inspection chamber to provide light, as shown in Figure 1.

Before the images acquisition system began, one group of the walnut kernels mixed with impurities was manually placed on the first-step conveyor belt, and the driving motor was started. After entering the second stage of the conveyor belt, the sample is paved. The walnut image was captured by the camera and stored in the computer. The image acquisition frequency was 2/s. About 130 images of walnut mixed with different impurities can be obtained by each group of the sample.

#### 2.2.2. Dataset Production

In order to improve the effectiveness of training and increase the diversity of samples, the collected image data were screened before training, and the images with low definition were removed. Finally, 1320 walnut kernel images were obtained and stored in JPG format. After processing by Matlab, the image resolution was set to 512 pixels × 512 pixels. In this paper, the dataset is enhanced by changing the adaptive contrast, rotation, translation, cropping and other methods, and the dataset is expanded to 5732 images [21]. The dataset contains four categories of labels: walnut shell, small impurities (diameter less than 5 mm), foreign impurities and metamorphic walnut kernels, as shown in Figure 2. The gray value range of the walnut kernel is the basis for identifying the deterioration degree of walnut kernels. All the images of walnut kernels are gray processed, and the gray value range of the metamorphic walnut kernel is from 20 to 35 after testing and statistics. The image labeling software is Labelimg, which is used to label the real bounding box and categories [22]. Then, according to the ratio of 3:1:1, all the enhanced images are divided into the training set, validation set and test set. There are 3439 images in the training set, 1146 images in the validation set and 1146 images in the test set.

#### 2.2.3. Experimental Equipment

The training of this model is conducted based on the Windows 10 operating system and the Pytorch framework. The CPU model of the test equipment is Intel^®^Core™ i7\11800H CPU@3.70 GHz, the GPU model is GeForce RTX 3080 10 G, and the software environment is CUDA 11.3, CUDNN 7.6 and Python3.8. The original YOLOv5 and the im-proved YOLOv5 are trained separately. The specific parameters are presented in Table 1.

### 2.3. Walnut Kernel Impurity Detection Based on YOLOv5

Currently, the target detection algorithms applied in food detection have high recognition accuracy, but the detection models often have too many parameters and large volumes, and are too complex and challenging to meet the needs of real-time detection [23]. Since the actual application site of walnut impurity detection is located in the food assembly line, the detection model should not only meet the requirements of recognition accuracy but also meet the real-time requirements of detection. YOLOv5 has a higher detection accuracy and a lighter model volume, so it has a faster response speed. Therefore, this paper adopts the YOLOv5 model for the detection of walnut impurities; its frame is shown in Figure 3.

### 2.4. Walnut Kernel Impurity Detection Based on YOLOv5

#### 2.4.1. Small Object Recognition Layer

There are small impurities, such as broken shells in the walnut images, and the detecting model used must be able to detect small objects. In the process of using the original YOLOv5 model, impurities such as the small broken shells of walnut kernels are small. The feature map in the YOLOv5 network structure is too small, while the multiple of the downsampling is large; thus, it is difficult for the deeper feature map to learn the features of small targets’ information, which lead to omissions of small impurities. To solve this problem, this paper tries to add a small object detection layer to the original YOLOv5 head, which will continue to process the feature map for expansion. After the 17th layer of the head part, it performs upsampling and other processing on the feature map so that the feature map continues to expand. At the 20th layer, the acquired feature map with a size of 160 × 160 is concated with the feature map of the second layer in the backbone to obtain a larger feature map for small target detection.

As shown in Figure 4, the function of upsampling is to enlarge the feature map so that the displayed image has a higher resolution, which is more conducive to detecting and recognising small targets. The upsampling process in this paper is implemented by the method of transposed convolution. Unlike the ordinary convolution, transposed convolution is adding a unit-step null pixel between each two pixels of the input image, so that the obtained Feature Map size becomes larger.

#### 2.4.2. Trans-E Block

The Transformer was first used in the field of natural language machine translation, and its most significant feature is the self-Attention mechanism. The main working modules in the Transformer structure are the encoder and decoder. During machine translation, the encoder part models the input sequence. It extracts the output value of the last time step at the structural output as a representation of the input sequence. The decoder then takes the input sequence representation as its input value and generates the translation with maximum probability. This paper simulates the encoder function in the Transformer structure, and proposes a Transformer-Encoder (Trans-E) block and tries to apply it to the image impurity detection. The structure of the Trans-E block is shown in the Figure 5. The Trans-E block consists of two sub-layers, the multi-head attention layer and the fully-connected layer. Among them, the multi-head attention layer is to perform multiple linear mappings of different sub-region representation spaces through multiple heads under the consideration of parallel computing; thus, it can obtain more comprehensive information under different sub-spaces at different locations. The main function of theconnected layer is to map the feature space calculated by the previous layer to the sample label space. A residual structure connects the two sub-layers. This article replaces the bottleneck blocks and some Cnov blocks of CSPDarknet53 in the original YOLOv5 with Trans-E blocks. Compared with CSP bottleneck blocks, Trans-E blocks have more advantages in capturing global information.

#### 2.4.3. CBAM Attention Mechanism

Since there is much useless information in the walnut kernel image, such as the walnut kernel itself, in order to suppress other useless image information, we increase the effective image feature weight, reduce the invalid weight, and make the training network model produce the best results. This paper introduces the based YOLOv5 Convolutional Block Attention Module (CBAM). The working principle of this module is as follows: take the global max pooling and global average pooling operations based on width and height, respectively for the input feature map *F* (*H* × *W* × *C*), and the output result is two 1 × 1 × *C* feature maps. Then, the obtained feature maps are sent to the neural network (*MLP*), respectively. The number of layers in the neural network is two layers. The number of neurons in the first layer is *C*/r (r is the reduction rate), the activation function is Relu, and the second layer is the number of neurons. The number of neurons in the layer is *C*. Then, an element-wise-based sum operation is performed on the output features, and the final channel attention feature, namely *M_c_*, is generated after the sigmoid activation operation. Finally, the element-wise multiplication operation is performed on *M_c_* and the input feature map *F* to generate the input features required by the Spatial attention module. The specific calculation is as follows:(1)$$M_{C}\left(F\right)=\sigma \left(MLP\left(AvgPool\left(F\right)\right)+MLP\left(MaxPool\left(F\right)\right)\right)\;\;\;\;\;\;\;\;\;\;\;\;\;\;\;\;\;\;\;$$
(2)$$=\sigma \left(W_{1}\left(W_{0}\left(F_{avg}^{c}\right)\right)+W_{1}\left(W_{0}\left(F_{max}^{c}\right)\right)\right)\;\;$$

The output of the channel attention module is taken as an input into the spatial attention module, which is also subjected to maximum pooling and average pooling. Then the two are stacked through the Concat operation, which only compresses the channel dimension but not the spatial dimension to focus on the target’s location information. The mechanism is shown in Figure 6.

In this paper, the CBAM module is added after the C3 module and the Trans-E module in the neck part so that the image features of walnut shells and foreign objects are weighted and combined, which increases the network at the cost of a small amount of computation, so that the network pays more attention to the key information of foreign objects such as walnut shells, which helps to train a better network.

#### 2.4.4. Ghostconv Makes Models Lightweight

Since the main part of the original YOLOv5 adopts the C3 structure for feature extraction, after adding the small target detection layer, the Trans-E block and the CBAM module based on the original network, the overall network has a large number of parameters. When the detection rate is low, it will be difficult to meet the real-time detection requirements. The actual scene of the walnut kernel impurity detection is a moving conveyor belt, so the detection model must have a relatively lightweight model and low detection delay. This paper applies the GhostConv block in GhostNet and replaces some ordinary convolution block in the current network model to make the detection model more lightweight.

Different from traditional convolution blocks, GhostConv performs feature map extraction on images in two steps [24]. The first step is still using the normal convolution calculation, and the feature map channel obtained at this time is less. The second step uses cheap operation (depthwise conv) to perform feature extraction again to obtain more feature maps, and then concat the feature maps obtained twice to form a new output.

As can be observed from Figure 7, the cheap operation will perform cheap computations on each channel to enhance feature acquisition and increase the number of channels. This mode requires significantly less computation than traditional convolution computations.

In order to solve the problem that the original YOLOv5 network cannot detect small impurities well and the detecting accuracy of individual near-color foreign objects is low, this paper combines the small object detection layer, Trans-E block, CBAM module and GhostConv to construct the entire improved YOLOv5 network model framework, as shown in Figure 8.

### 2.5. Experiment Process

First, the manual labeling method is used to mark each walnut image to obtain the training label image, and then the walnut image set is divided into training set, validation set and test set according to the ratio of 3:1:1. The training set is input into the improved YOLOv5 network for training. During the training process, the stochastic gradient descent algorithm is used to optimize the network model, and the optimal network weights are obtained when the training is completed. Subsequently, the images in the validation set of weight values are used to test the performance of the network model and compare with the test results of the original YOLOv5 model and other prediction models. The feasibility of the walnut kernel impurity detection model based on the improved YOLOv5 was verified. The test process is shown in Figure 9.

### 2.6. Model Evaluation Index

The model loss of YOLOv5 consists of bounding box loss, object loss and classification loss, which can be used to test the target prediction performance of the model. Precision (Pre) and recall (Rec) can intuitively reflect the accuracy of target prediction, which are calculated by the ratio of the number of *TP*, *FP*, *TN*, and *FN* [25], where *TP* represents the number of correctly detected positive samples, and FP represents the error Number of negative samples detected, *FN* indicates the number of positive samples not detected. The *F*1 score is the weighted average of precision and recall. The AP value of each class is the area composed of the label P-R map of that class. The mean average precision (*mAP*) is the average of the *AP* values of various labels; thus, it can represent the global detection performance of the model.
(3)$$\;loss=l_{bbox}+l_{object}+l_{classification}$$
(4)$$Pre=\frac{TP}{\left(TP+FP\right)}$$
(5)$$Rec=\frac{TP}{\left(TP+FN\right)}$$
(6)$$F_{1}=\frac{2\times Pre\times Rec}{Pre+Rec}$$
(7)$$\;AP=\int_{0}^{1}Pre\left(Rec\right)dRec$$
(8)$$mAP=\frac{1}{\left|Q_{R}\right|}\underset{q=Q_{R}}{\sum }AP\left(q\right)$$

## 3. Results and Discussion

### 3.1. Model Training Results

According to the data set type, the loss function of the prediction model can be divided into training loss and validation loss, and the curve is shown in the Figure 10a. It can be observed from the figure that in the process of model training, when the number of iterations is between 0 and 150, the training loss and validation loss decrease rapidly, and when the number of iterations reaches more than 250, the loss value of the prediction model begins to stabilize gradually. In this paper, the training model with 300 iterations is selected as the final walnut kernel impurity detection model. In addition, it can be observed from the *mAP* curves of the training set and the validation set in the Figure 10b that the trained prediction model does not appear to be overfitting.

### 3.2. Model Test Results and Analysis

In order to verify the performance of the detection model, the number of impurities for each category in the random 300 images in the validation set was counted and calculated, and then compared with the test results of the model. Among the 300 images in the validation set, the number of walnut shell impurities is 2059, the number of metamorphic walnut kernels is 786, the number of small impurities is 2621, and the number of other impurities is 432. The precision rate, recall rate, *F*1 score and *mAP* were used to evaluate the prediction accuracy of the model for various impurities. The predicted results are shown in Table 2.

The confusion matrix can intuitively reflect the prediction results of classification problems, showing the prediction probability for each category. From the confusion matrix in the Figure 11, it can be observed that among the four types of impurities, the detection accuracy of the walnut shell is the highest, which can reach 92.21%, and the detection accuracy of small impurities is the lowest. Since impurities are located at the boundary of the image, the annotation information is accurate, resulting in a small part of the spoiled walnut kernels being predicted as other impurities.

### 3.3. Performance Comparison of Different Models

In order to better verify the performance of the improved walnut kernel impurity detection model, 300 images in the above validation set were used as the test objects, and the original YOLOv5, YOLOv4, Faster R-CNN, and SSD300 models were used to test and compare the test results [26]. Similarly, the accuracy, *F*1 score and *mAP* are used as indicators to evaluate the performance of the model. Considering that in actual nut processing, the detection rate of the model is high to meet the needs of real-time detection, so it is also necessary to use the model size and the average GPU detection speed as the evaluation indicators of the model. The test results of each model are shown in Table 3.

As can be observed from the data in the Figure 12, the detection accuracy of the improved YOLOv5 model is 5.77% higher than that of the original YOLOv5, and both are higher than other detection models, *mAP* has increased by 6.79%, and $F_{1}$ has increased by 5.06%. The result is also better than the fire inspection small target detection model based on YOLO algorithm, whose *mAP* is 80.23% and $F_{1}$ is 73% [27]. The experiment proves that the introduction of the small target detection layer, the replacement of the Trans-E block, and the introduction of the CBAM module on the basis of the original YOLOv5 model can help improve the accuracy and performance of walnut kernel impurity detection.

Model detection speed is also one of the important performance indicators for real-time detection of food impurities. While improving the accuracy of impurity detection, the YOLOv5 model parameters have increased, and the model size has also increased by 1.74 M. At the same time, the detection time of a single image is increased to 65.25 ms, which is 21.51 ms longer than the original YOLOv5 single image detection time. In order to reduce the detection time of a single image and improve the efficiency of real-time detection of impurities, this paper replaces the conventional Conv of the main part and the detection head part with Ghostconv to make the model more lightweight. After replacing Conv with Ghostconv, the single image impurity detection time is reduced from 65.25 ms to 45.38 ms, which is only 4.99% longer than the original YOLOv5 detection time. Compared with the improved SE-YOLOv5, the detection response time is reduced by 10.4%. [17] This model also leads to other commonly used detection models such as YOLOv4 in the detection rate performance. Therefore, the improved YOLOv5-based walnut kernel impurity modeling model is a suitable detection model.

### 3.4. Comparison of Recognition Result

Figure 13 compares the results of the original YOLOv5 and the improved YOLOv5 for detecting impurities in walnut kernels. The brown boxes in the figure represent walnut shells, the green boxes represent mildewed walnut kernels, and the red boxes represent small impurities. It can be observed from the figure that the missed detection rate of small impurities in the improved YOLOv5 model is greatly reduced, and the corresponding target confidence is improved. Under the background of high-density walnut kernels and extremely small impurities, the original YOLOv5 has a weak ability to extract features, resulting in the inability to accurately predict the impurity target. The detection performance of the improved YOLOv5 model is significantly better than the former, with a large number of detected small targets and high accuracy, and better performance in detecting small impurity targets.

## 4. Conclusions and Future Research

The detection of walnut impurities is of great significance to the safety of nut food. In this paper, an impurity detection model of walnut kernels based on the improved YOLOv5 network is established: a small target recognition layer is added to the original prediction head of the model to obtain more small impurities feature information. Then, some convolution blocks in the network are replaced by Trans-E blocks, which can capture more comprehensive information in different subspaces at different locations. The CBAM attention module is added to the neck part of the network model for feature fusion, which improves the network performance at a small cost. Finally, Ghostconv is introduced to replace the original Conv, which reduces the computational burden of the model and improves the detection speed. The improved model detection *mAP* can reach 88.9% and *F*1 can reach 90.81%, which is better than the original YOLOv5 network and other networks. Moreover, the improved network model has not only a high detection rate, but also a significant improvement in the identification rate of small target impurities. The model improvement studied in this paper is to maintain a balance between detection performance and detection speed, so as to meet the demand of the real-time detection of walnut impurities. Near infrared spectroscopy is an important tool in the field of food impurity detection [28]. However, it requires demanding hardware. The detection technology based on YOLOv5 has a higher detection rate, lighter detection equipment and a wider range of application objects when compared to the near infrared spectroscopy. It also has certain advantages in detection accuracy. The research content is also applicable to other nut food impurity detection fields, and provides technical reference for the detection of snack food impurities.

However, the improved YOLOv5 model has limitations, such as a fraction of missing and wrong detection cases for small foreign bodies. Therefore, the detection accuracy of the model still needs to be improved. Improving the resolution of the camera is conducive to improving the detection accuracy. Then, due to the influence of external light source, the illumination of the image is biased. Fan Youchen et al. improved the YOLOv5 combined with dark channel enhancement to solve the problem of insufficient illumination. [29] This method can be applied to solve the illumination problem of the image. In addition, making the detection model lighter is one of the key points of future research. Chu et al. proposed a real-time apple flower detection method based on YOLOv4 and using the channel pruning method. [30] Isa Iza Sazanita et al. used the adaptive moment estimation optimizer and the function reducing-learning-rate-on-plateau to optimize the model’s training scheme [31]. In the future, we can try to replace the backbone network with other lightweight networks to reduce the number of model parameters.

## Figures and Tables

**Figure 1 foods-12-00624-f001:**
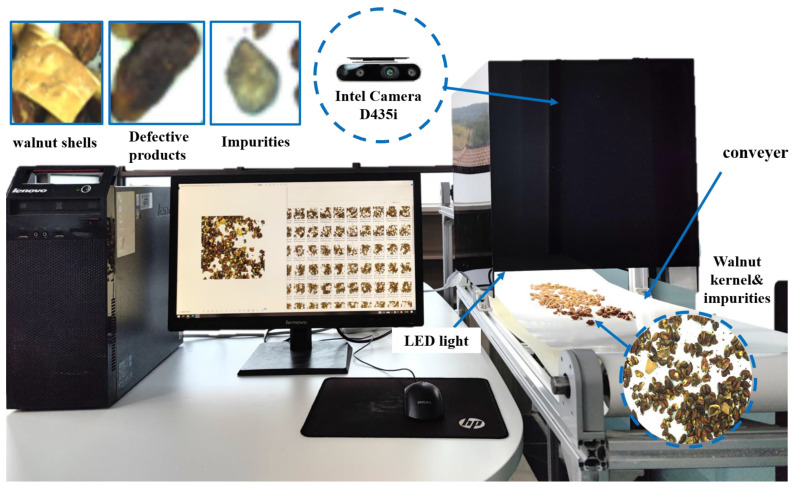
The hardware of the image acquisition system in the lab.

**Figure 2 foods-12-00624-f002:**
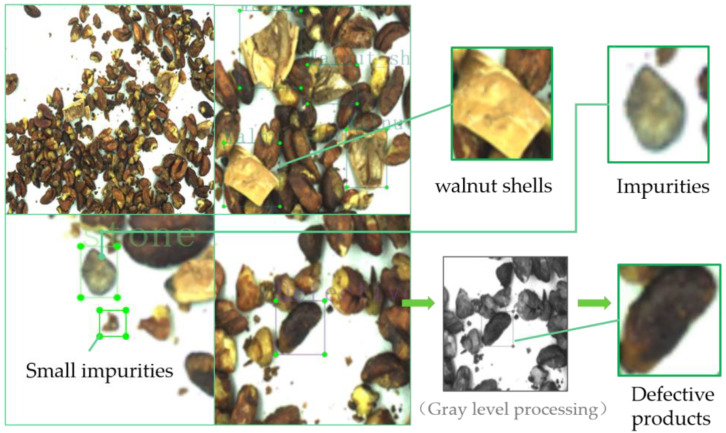
Walnut kernel impurity type labeling.

**Figure 3 foods-12-00624-f003:**
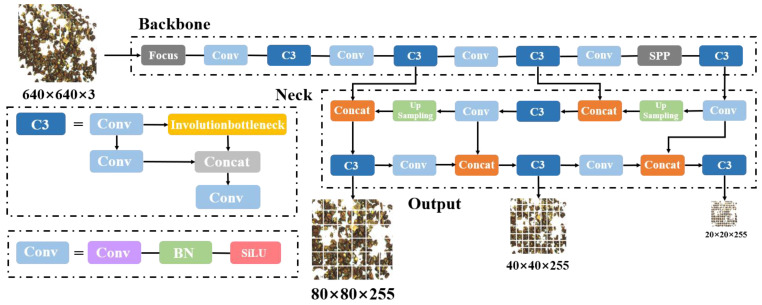
The impurity detection model of walnut kernel based on YOLOv5.

**Figure 4 foods-12-00624-f004:**
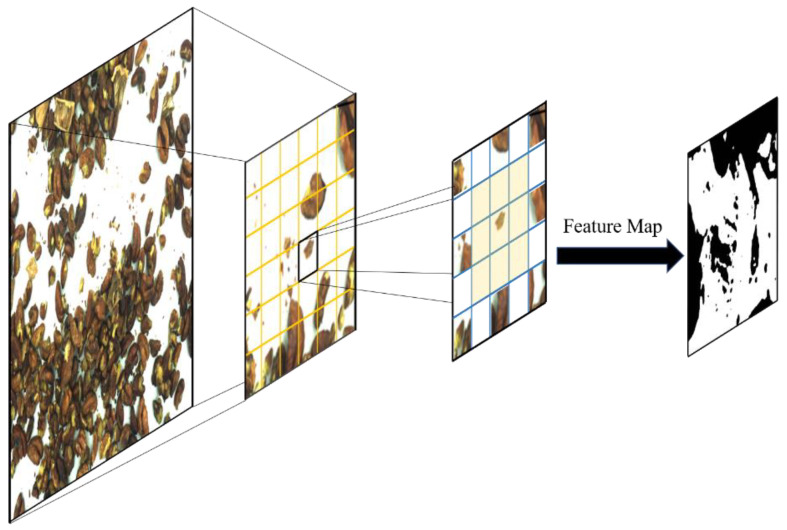
Schematic diagram of upsampling.

**Figure 5 foods-12-00624-f005:**
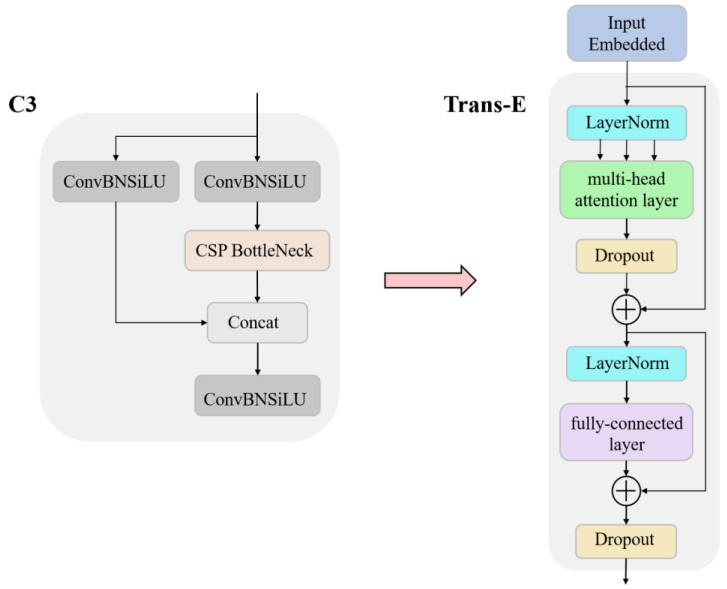
The architecture of Tran-E block.

**Figure 6 foods-12-00624-f006:**
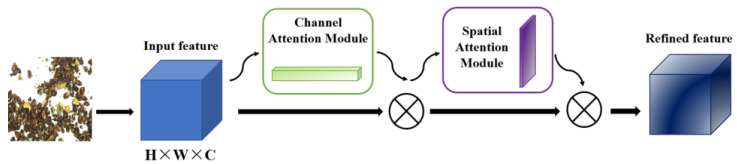
CBAM attention mechanism.

**Figure 7 foods-12-00624-f007:**
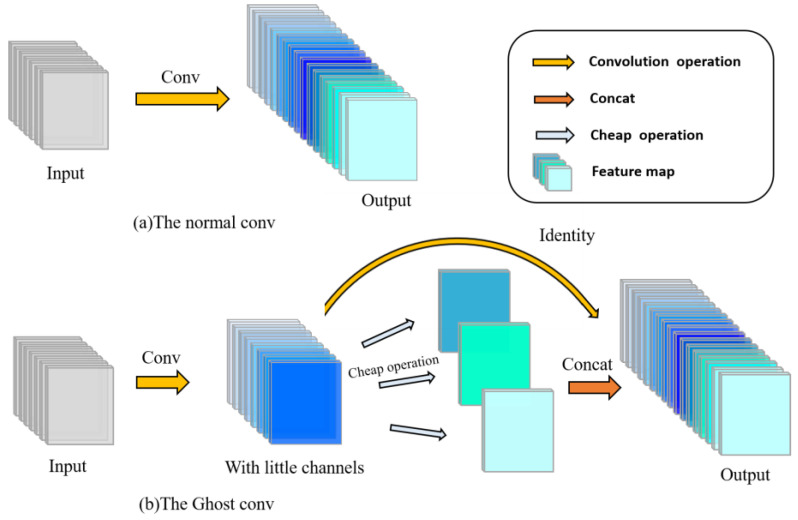
(**a**)The ordinary convolution. (**b**)The Ghost convolution.

**Figure 8 foods-12-00624-f008:**
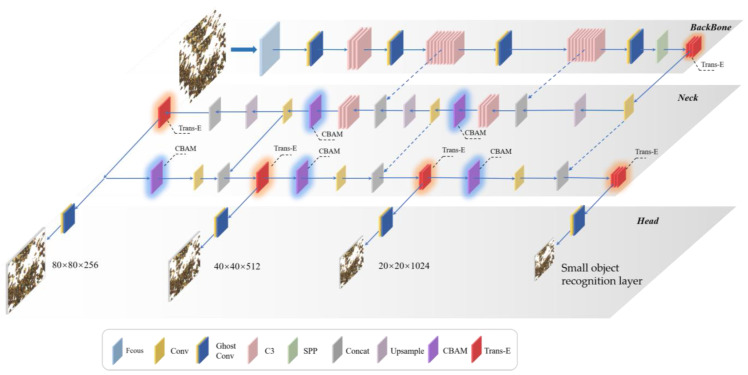
Walnut kernel impurity detection model based on improved YOLOv5 network.

**Figure 9 foods-12-00624-f009:**
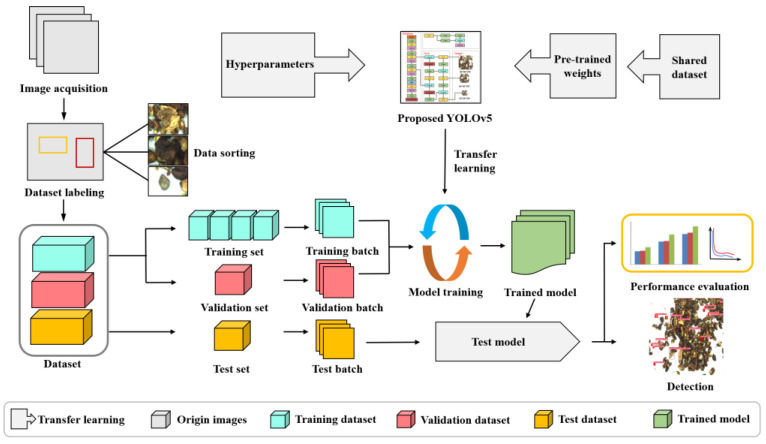
Flowchart of the overall workflow methodology for the proposed detection model.

**Figure 10 foods-12-00624-f010:**
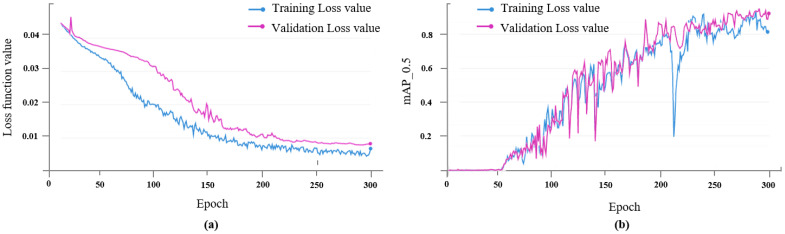
Training results of the improved YOLOv5 model. (**a**) Training and validation loss. (**b**) *mAP*_0.5 of training and validation sets. *mAP*_0.5: mean average precision when the threshold of IoU is 0.5.

**Figure 11 foods-12-00624-f011:**
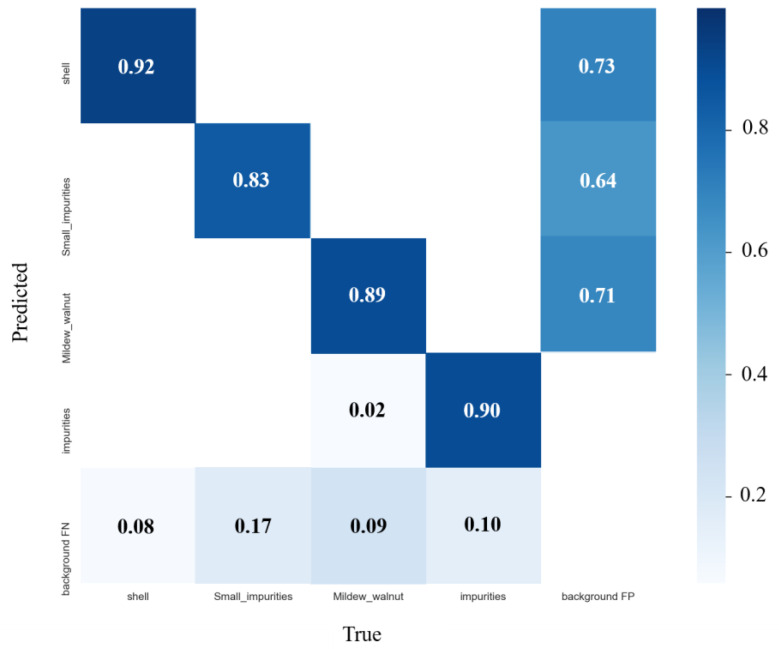
Confusion matrix of four kinds of impurities.

**Figure 12 foods-12-00624-f012:**
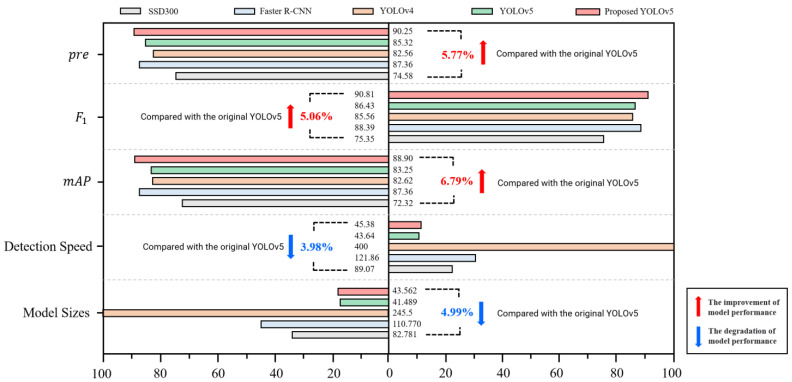
Comparison of the performance from different network models.

**Figure 13 foods-12-00624-f013:**
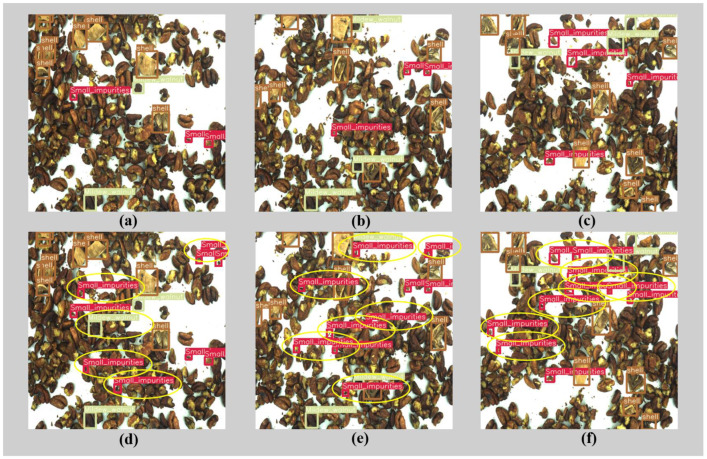
Comparison of the recognition effects of the YOLOv5 models. (**a**–**c**) recognition effects of the original YOLOv5; (**d**–**f**) proposed YOLOv5 network.

**Table 1 foods-12-00624-t001:** Test environment setting and parameters.

Parameter	Configuration
Operating system	Windows 10
Deep Learning Framework	Pytorch2.6
Programming language	Python3.8
GPU accelerated environment	CUDA 11.3
GPU	GeForce RTX 3080 10 G
CPU	Intel^®^Core™ i7\11800H CPU@3.70 GHz

**Table 2 foods-12-00624-t002:** Recognition results of targets using improved YOLOv5 model.

Class	Num	*Pre* (%)	Rec (%)	*mAP* (%)	*F*1 (%)
Shell	2059	92.21	96.32	94.20	94.56
Small_impurities	2624	83.56	87.84	85.12	86.21
metamorphic_walnut	786	89.24	93.37	90.98	91.26
Other impurities	432	90.25	94.93	92.21	92.87
Total	5901	89.69	93.42	91.25	91.77

**Table 3 foods-12-00624-t003:** Comparison of precision, recall, F1-score, mean Average Precision, detection speed and ModelSizes between proposed model and other advanced models.

Model	P (%)	R (%)	*F*1-Score (%)	*mAP* (%)	Dect. Time (ms)	ModelSizes (M)
Faster-RCNN	87.36	89.25	88.39	81.62	121.86	110.770
SSD300	67.75	75.38	65.43	69.36	89.07	82.781
YOLOv4	82.56	90.14	85.56	85.62	400	245.5
YOLOv5	85.32	88.97	86.43	83.25	43.64	41.489
Proposed YOLOv5	90.25	91.56	90.81	88.9	45.38	43.562

## Data Availability

The data are available from the corresponding author.

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
