# Peer review of "An Improved YOLOv5 Model: Application to Mixed Impurities Detection for Walnut Kernels"

_foods, 2023, doi:10.3390/foods12030624_

Round 1
Reviewer 1 Report
The submitted article contains sufficient novelty and deserves readers attention. However, several things needs to be improved.
- Abstract can be improved.
Instead of : Impurity detection is an important link in the process of food processing, better option is : Impurity detection is an important link in the chain of food processing. (avoid using process and processing in the same sentence).
Instead of : In order to accurately identify the small impurities mixed in walnut kernels, this paper established an improved impurities detection model based on the original YOLOv5 network model: a small target detection layer was added in the Neck part to improve the detection ability for small impurities like broken shells. Secondly, a Tans-E module is proposed to replace some convolution blocks in the original network, which can better capture the global information of the image. Then add the CBAM module to improve the sensitivity of the model to channel features, which make it easy to find the prediction region in dense objects.
Should be : In order to accurately identify the small impurities mixed in walnut kernels, this paper established an improved impurities detection model based on the original YOLOv5 network model. Initially, a small target detection layer was added in the Neck part, to improve the detection ability for small impurities like broken shells. Secondly, a Tans-E module is proposed to replace some convolution blocks in the original network, which can better capture the global information of the image. Then, the CBAM module was added to improve the sensitivity of the model to channel features, which make it easy to find the prediction region in dense objects.
Instead of : In the experimental stage, in order to verify the effectiveness of the model, sample images were randomly selected from the constructed walnut database to form the training set and the test set.
Should be : During the test stage, sample photos were randomly chosen to test the model's efficacy using the training and test set, derived from the walnut database that was previously created.
It would be preferable to avoid extensive use of abbreviations in the abstract, because abstract is standalone part of the manuscript. At least some abbreviations should be defined or avoided (like mAP).
- Keywords.
“an improved model” cannot be a keyword. Authors should find another one.
- Unusual citing of references.
The authors have chosen to cite all the references in introduction. No references were mentioned in discussion. The references portfolio must be expanded and the results of the manuscript must be compared with previously published data in the literature.
- References list.
References are sloppy written! The authors are urged to devote special attention to this part of manuscript. Reference authors should be cited by their last name in the text. Also, reference list must be formatted according to Journal instructions. List of cited articles cannot contain et. al. expressions!
- Page 10, line 284, the authors refer to a figure but it is not stated which figure.
Author Response
Thank you for your sincere advice!
We have responded to your comments point-by-point.
Please see the attachment.

Reviewer 2 Report
My correction with the manuscript are relatively minor, and are listed below. I can recommend these corrections. The topic of the study is interesting and it fits into trends in science as well as in the manufacturing practice.
· The abstract explains the purpose of the work and includes background information. It is also necessary to describe in a few simple steps the process of processing and obtaining results and the reason for choosing these parameters.
· The introduction provides a good general background on the subject and gives the reader an idea of the wide range of possible applications of this technology.
· The methods used in this paper are appropriate for the purpose of the study.
Lines 104-124: Please provide a reference for this method.
Line 105: The proposal is to replace the term "light source black box" with something more appropriate.
Line 108-109: Data on photo quality, color model, etc. are missing. The way the methods are currently written does not allow other researchers to replicate the experiment (because they are missing important details).
· Results: Some deficiencies were noted in the presentation of the results. Many titles of figures/tables do not contain enough information to easily follow the text. Abbreviations should be avoided when naming figures, especially if they are not explained in the title. When looking at the results, the reader should follow the presented results independently, without paying attention to the abbreviations in the rest of the text.
· Conclusions presented in this paper correlate to the results found. Given the scope of the results presented please describe research limitations for future research.
Author Response
Thanks for your valuable advice!
We have responded to your comments in the word point-by-point.
Please see the attachment.
